# Association with Ambient Air Pollutants and School Absence Due to Sickness in Schoolchildren: A Case-Crossover Study in a Provincial Town of Japan

**DOI:** 10.3390/ijerph18126631

**Published:** 2021-06-20

**Authors:** Masanari Watanabe, Hisashi Noma, Jun Kurai, Kazuhiro Kato, Hiroyuki Sano

**Affiliations:** 1Division of Internal Medicine, Motomachi Hospital, 1895-1 Agarimichi-cho, Sakaiminato 684-0033, Japan; junkurajun@gmail.com; 2Department of Data Science, Institute of Statistical Mathematics, 10-3 Midori-cho, Tachikawa, Tokyo 190-8562, Japan; noma@ism.ac.jp; 3Department of Internal Medicine, Yasugi Daiichi Hospital, Yasugicho 899-1, Yasugi 692-0011, Japan; eugene.aleuc@gmail.com; 4Allergy Center, Kindai University Hospital, Ohnohigashi 377-2, Osakasayama 589-0014, Japan; hsano@med.kindai.ac.jp

**Keywords:** ambient air pollutants, Asian dust, cough, fever, school absence, sickness

## Abstract

The effect of ambient air pollutants and Asian dust (AD) on absence from school due to sickness has not been well researched. By conducting a case-crossover study, this study investigated the influence of ambient air pollutants and desert sand dust particles from East Asia on absence from school due to sickness. From November 2016 to July 2018, the daily cases of absence due to sickness were recorded in five elementary schools in Matsue, Japan. During the study period, a total of 16,915 absence cases were recorded, which included 4865 fever cases and 2458 cough cases. The relative risk of overall absence in a 10-μg/m^3^ increment of PM_2.5_ and a 0.1-km^−1^ of desert sand dust particles from East Asia were found with 1.28 (95%CI: 1.15–1.42) and 2.15 (1.04–4.45) at lag0, respectively. The significant influence of PM_2.5_ persisted at lag5 and that of desert sand dust particles at lag2. NO_2_ had statistically significant effects at lag2, lag3, and lag4. However, there was no evidence of a positive association of O_x_ and SO_2_ with absence from school. These results suggested that PM_2.5_, NO_2_, and AD increased the risk of absence due to sickness in schoolchildren.

## 1. Introduction

As the world gets hotter and more crowded, ambient air pollutants are considered the largest environmental health threat globally [1]. Low- and middle-income countries disproportionately experience this burden compared to the developed countries [2]. Mainland East Asia has undergone rapid industrial development in the recent decades, which has increased ambient air pollutants in the boundary layer and led to this region now being one of the most heavily polluted areas in the world [3,4,5,6]. The transboundary transport of ambient air pollutants from the East Asian continent to Japan has contributed to an increase in air pollution levels [7]. Although particulate matter (PM) pollution level has been slowly improving in recent years following the environmental policy by the Chinese government, the average annual level of ozone in Japan remains on an upward trend [7]. Thus, even in developed East Asian countries such as Japan, China, and South Korea, ambient air pollutants continue to be an important health concern.

Among ambient air pollutants, PM, ozone, nitrogen dioxide (NO_2_), and sulphur dioxide (SO_2_) have the strongest evidence for causing various health disorder [8,9,10]. These have been responsible for 4.2 million deaths globally and are estimated to cause about 16% of the lung cancer deaths, 25% of chronic obstructive pulmonary disease (COPD) deaths, about 17% of ischemic heart disease and stroke, and about 26% of respiratory infection deaths around the world [1,11]. In children, ambient air pollutants are an important risk factor for acute and chronic diseases such as respiratory diseases, allergic diseases, cancers, and acute respiratory infections [1,12,13,14,15,16,17]. Furthermore, they are associated with increased deaths, hospitalisation, and emergency visits. Several studies surveyed the association between ambient air pollutants and school absence [18,19,20,21,22,23] and found that PM less than 2.5 μm aerodynamic diameters (PM_2.5_) and ozone were associated with school absence [18,19,20,21,22]. Recent studies showed that ambient air pollutants control and environmental regulations can play significant inhibitory roles in their emissions [24] and can reduce adverse health effects [25,26].

Asian dust (AD), originating in the deserts of Mongolia, northern China, and Kazakhstan and often dispersing over East Asia, is also an important health concern in East Asia. Originally, AD was a desert sand dust storm; however, with rapid industrial development in East Asia it has become an important source of ambient air pollutants [27]. Studies have shown that AD has an influence on mortality, emergency visits, and hospitalisations for diseases such as cardiovascular disease, pneumonia, and asthma in both children and adults [28,29,30,31,32].

Numerous studies found adverse effects of ambient air pollutants and AD on respiratory and cardiovascular diseases and investigated the relationship with a focus on admissions, emergency visits, and hospital visits. However, to the best of our knowledge, there are few reports about the influence of AD on absence from school due to sickness in schoolchildren. Although a few studies have found an association between them, but they did not investigate the specific causes of the absence [18,19]. Thus, many points, specifically, the potential for such absence need further clarification. Children’s exposure to ambient air pollutants is a special concern because, when the exposure begins, their immune system and lungs are not fully developed, raising the possibility of different responses than that observed among adults [33]. The aim of this study, therefore, was to investigate the influence of short-term exposure to ambient air pollutants and AD on absence from school due to sickness among Japanese schoolchildren. Furthermore, this study examined the symptoms to identify the effect of infection and respiratory disorder on absence due to exposure to ambient air pollutants and AD.

## 2. Materials and Methods

### 2.1. Study Design

This study focused on the influence of infection and respiratory disease exposure to ambient air pollutants and desert sand dust particles on absence from school. It used a case-crossover design, which compared the exposure on the case day, that is, when events occurred, with referent days and examined the differences in exposure that contributed to the differences in the daily count of cases. From 1 November 2016 to 20 July 2018, the number of absences due to sickness were recorded in five elementary schools of Matsue City in western Japan. The sickness details were also recorded and divided into eight categories: fever up, headache, cough (respiratory symptom), diarrhoea and abdominal pain (abdominal symptoms), vomiting, rashes, influenza infection, influenza like symptoms defined as having both high fever and upper airway symptoms, and other reasons. These classifications decreased the efforts required by schoolteachers and made taking the survey easier. Multiple answers were accepted as reasons for the absence. Matsue City has a population of approximately 200,000 individuals and covers an area of 530.2 km². The five elementary schools were within 5 km of each other, and the students lived within a 1 km radius of the schools. The study was approved by the Ethics Committee of the Faculty of Medicine, Tottori University (approval number 1607A033).

### 2.2. Measurements of The Levels of Air Pollutants, Sand Dust Particles, and Weather Data

The hourly concentrations of PM_2.5_, SO_2_, NO_2_, and photochemical oxidants (O_x_) are monitored by the Japanese Ministry of the Environment in Matsue City, and these data were obtained from the ministry database. The data of meteorological variables, including daily temperature, humidity, and atmospheric pressure, were obtained from the Japan Meteorological Agency.

Light Detection and Ranging (LIDAR) systems measure PM as non-spherical airborne particles, which are equivalent to desert sand dust particles from East Asia; spherical airborne particles, which are equal to air pollution aerosols, are measured by illuminating a target with two laser beams of different wavelengths and analyzing the reflected light [34,35]. Both levels of spherical and non-spherical particles are measured at 15-min intervals, and daily particle levels are determined from the median value of 96 measurements collected over a 24-h period from midnight of the first day to midnight of the following day. LIDAR is used to monitor the concentration of desert sand dust particles from East Asia. For the current analysis, LIDAR data for spherical and non-spherical particles were obtained from the Matsue observatory. Values measured at 120–150 m above the ground, which is the minimum altitude required by LIDAR systems to measure non-spherical and spherical particles, were used.

### 2.3. Statistical Methods

The data set of this study was constructed to facilitate a time-stratified case-crossover analysis [36,37,38], in which each case acted as its own control. This study adopted the conditional Poisson model to assess associations between time series of environmental exposures and counts of health outcomes with adjustment for overdispersion and autocorrelation. Average temperature, relative humidity, and atmospheric pressure were involved in the Poisson regression model to adjust the potential confounding. In order to combat the potential delayed effects of ambient air pollutants on absence due to sickness, this study used the single-day lags (lag0, lag1, lag2, lag3, lag4, lag5) to estimate the effects of air pollutants at different lag days. Lag1 referred to the concentration of air pollutants on the previous day. The relative risk (RR) and its 95% confidence interval (CI) for a 10 µg/m^3^ increase in 24-h average concentration of PM_2.5_, a 1-ppb increase in 24-h average concentration of NO_2_, SO_2_, and O_x_ and a 0.1-km^−1^ increment in desert sand dust particles by LIDAR were calculated. A value of 0.1 km^−1^ in the concentration of desert sand dust particles from East Asia by LIDAR corresponded to 0.1 mg/m^3^ [39]. Statistical significance was considered only when the *p*-value was smaller than 0.05.

## 3. Results

The spatial distributions of schools are displayed in Figure 1. All of them were concentrated in the main urban areas of Matsue City. During the study period from 1 November 2016 to 20 July 2018, there were 361 school days in 627 days. The descriptive statistics for the meteorological factors and air pollutants during the study period are summarised in Table 1. Data are shown as mean, median, and quartile.

Table 2 shows the results of the spearman rank correlation between average levels of air pollutants and meteorological factors. PM_2.5_, desert sand dust particles, and SO_2_ had significant relationships with NO_2_ and O_x_. A significant correlation coefficient between NO_2_ and O_x_ was not observed. SO_2_, NO_2_, and desert sand dust particles had significantly negative correlations with temperature, while there was a significantly positive relationship between PM_2.5_ and temperature.

During the study period, there were 266 school holidays in 627 days. Table 3 shows the total number of absences from school during the study period and due to fever and respiratory symptoms. The study included a total of 16,915 absence cases from school, of which 4865 were for fever and 2458 were for cough (as a respiratory symptom). The most common cause was an influenza infection which resulted in 5266 absences. The other causes were 1441 for headache, 1742 for abdominal symptoms, 981 for vomiting, 187 for influenza-like symptoms, and 50 for rashes. One thousand four hundred fifty-six cases were other causes. Daily mean absence cases were 47.0 for total, of which 13.5 were for fever and 6.8 were for cough.

The relative risk of an influence of air pollutants and desert sand dust particles on the overall cases of absence from school by single-pollutant models at different lags are presented in Table 4. A 10-μg/m^3^ increment in 24 h average concentration of PM_2.5_ was found at lag0 with 1.28 (95%CI: 1.15–1.42) and a 0.1-km^−1^ increment in desert sand dust particles at lag0 with 2.15 (1.04–4.45). There was significant evidence of a positive association of PM_2.5_ and desert sand dust particles with overall cases of absence from school. These significant influences of PM_2.5_ on overall cases of absence persisted at lag5 and that of desert sand dust particles at lag2. Increased NO_2_ had a statistically significant effect at lag2 with 1.07 (1.02–1.13), lag3 with 1.08 (1.04–1.13), and lag4 with 1.04 (1.00–1.08). On the contrary, there was no evidence of a positive association of O_x_ and SO_2_ with overall absence from school. 

The influence of air pollutants and desert sand dust particles on absence from school due to fever and cough are shown in Table 5 and Table 6, respectively. There were significant associations between PM_2.5_ and absence from school due to fever from lag0 to lag1, and a significant influence of desert sand dust particles on absence from school due to fever had at lag0 and lag1. On the contrary, cough was significantly associated with PM_2.5_ at lag0 and lag1 and desert sand dust particles at lag0. NO_2_ had a statistically significant association with absence due to fever at lag3 and cough from lag3 to lag5. SO_2_ was significantly associated with respiratory symptom at lag0. There was no evidence of a positive association of O_x_ with both cough and fever.

The relative risks of influence of PM_2.5_ and desert sand dust particles on the overall cases of absence at lag1 were higher than at lag0. In order to adjust for other pollutants, a two-pollutant model analysis at lag1 was performed in Table 7. The relative risks for the overall cases of absence in PM_2.5_ and desert sand dust particles appeared to be statistically significant, which was similar to the results of the single-pollutant models. However, for absence due to fever, the relative risks in PM_2.5_ appeared to be statistically insignificant after adjusting for desert sand dust particles. On the contrary, desert sand dust particles were similar to the results of single-pollutant models. In the results of absence due to cough, the relative risks of PM_2.5_ tended to be statistically significant adjusting for O_x_ and desert sand dust particles. The relative risks of desert sand dust particles on absence due to cough appeared to be statistically insignificant after adjusting for other air pollutants.

Table 8 shows the relative risk for a 10-unit increment of each meteorological factor on the overall cases of absence from schools, depending on levels of temperature, relative humidity, and atmospheric pressure. For temperature and the overall cases of absence, the association was both negative and statistically significant with an estimated reduction in risk of 0.72 (0.56–0.92) at lag4 and 0.71 (0.56–0.91) at lag5 per 10 ℃ increase. A significant relationship between the overall cases of absence and relative humidity was found at lag5 with 1.08 (1.01–1.15) per 10% increase. The increase in atmospheric pressure had a statistically significant effect at lag3 with 1.21 (1.05–0.92) and lag4 with 1.21 (1.08–1.38).

## 4. Discussion

Previous studies have unmasked the deleterious effects of ambient air pollutants and AD on emergency visits and hospitalisation for diseases in children. This evidence would predict that ambient air pollutants and AD may increase absence from school due to sickness in schoolchildren. However, the epidemiologic evidence focusing on their effects on children’s absence from school remains limited. Therefore, the present study investigated the relationship between absence from school and air pollutants and desert sand dust particles from East Asia among schoolchildren. PM_2.5_, NO_2,_ and desert sand dust particles had statistically significant adverse effects on the overall absence cases from school due to sickness; they were also associated with cough and fever. These results suggest that exposure to PM_2.5_, NO_2_, and AD have an impact on absence in the western part of Japan. Important causes of absence induced by PM_2.5_, NO_2_, and AD, may be infections and respiratory disorders.

Respiratory tract infection and disease are common causes for hospitalisation and emergency visits due to exposure to ambient air pollutants. An accurate diagnosis was unable to be made because this study was based on the declaration from schoolchildren and their guardians. Therefore, we defined infection based on the presence of fever because fever is the most important symptom in infection. We made a diagnosis of respiratory disease in case the absence from school was due to cough, which was categorised as a respiratory symptom because cough is one of the most important symptoms in respiratory disease. PM_2.5_ and desert sand dust particles had significant associations with both cough and fever. Infection and respiratory disease, including acute respiratory disease and exacerbation of chronic respiratory disease such as asthma, may be important causes of absence from school due to exposure to PM_2.5_ and desert sand dust particles.

The emission of ambient air pollutants over neighbouring areas and foreign countries is important in association with ambient air pollutants and health [24,40]. The sum of the contribution from foreign anthropogenic sources is larger than domestic contribution in most areas of Japan, excluding the Kanto region, which is the Greater Tokyo Area encompassing seven prefectures [40]. This study found a significant relationship between desert sand dust particles and other ambient air pollutants, suggesting that foreign ambient air pollutants had impacted that area. Exposure to ambient air pollutants in children is a special concern because their immune system and lungs are not fully developed when the exposure begins, raising the possibility of different responses than that observed among adults [33]. Additionally, children spend more time outside, where the concentrations of pollution from traffic, powerplants, and other combustion sources are generally higher [33]. Although ambient air pollutants have long been thought to exacerbate minor acute illnesses, recent studies have suggested that such pollutants, particularly traffic-related pollution, are associated with infant mortality and the development of asthma and atopy [25,26,33,41]. Some 93% of children and teens younger than 15 years worldwide are exposed to ambient air pollutants with PM levels higher than the limits prescribed in the WHO air quality guidelines [42]. Improvements in air quality, which can be achieved through regional and international cooperation in environmental control and informal regulation, will be beneficial to children worldwide [24,26].

Although the present study also surveyed absence due to headache, abdominal symptoms, vomiting, rash, influenza infection, influenza-like symptoms, and others, its primary outcome focuses on the influence of infection and respiratory disease. Few studies have reported the influence of ambient air pollutants on digestive diseases. Hence, these classifications were included to reduce schoolteachers’ efforts on their routine survey. Accordingly, the current study did not focus on the relative risk of the influence of ambient air pollutants and desert sand dust particles on cases of absence from school for these symptoms.

LIDAR systems can measure the amount of AD particles transported for long distances from East Asia to Japan because they are simultaneously applied within <1 km above the ground, and the measurements are continuously obtained from various locations in Japan, South Korea, China, Mongolia, and Thailand [34,35]. Accordingly, the levels of desert sand dust particles (non-spherical airborne particles) measured by LIDAR measurements are equivalent to the quantity of AD particles. There was significant evidence of a positive association of desert sand dust particles with absence from school in the overall cases and cases of fever and acute respiratory symptoms. These findings suggest that AD is able to increase absence from school through deteriorating infection and respiratory disease.

As well as ambient air pollutants, meteorological factors such as temperature, humidity, and ambient air pollutants are a few of the potentially modifiable environmental risk factors that do not depend on the change of individual behaviours. Several studies reported that there were interactions between ambient air pollutants and temperature on emergency department visits and hospitalisation [43,44]. Therefore, average temperature, relative humidity, and atmospheric pressure were included in the model to control the confounding effects of meteorological factors on the association between ambient air pollutants, which included desert sand dust particles, and absence from school. The present study also analysed the effects of the meteorological factors on absence from school and found statistical significances. PM_2.5_ and desert sand dust particles were significantly associated with absence from school in lag0, lag1, and lag2, but the significant relative risk for meteorological factors on absence from school lacked in lag0, lag1, and lag2. Meteorological factors did not affect interactions between ambient air pollutants, desert sand dust particles, and absence from school due to sickness.

The present study did not include other meteorological factors like rain and wind speed into the models, because there is little evidence supporting the associations between them and getting sick. The influence of the daily amount of precipitation and wind speed on absence from school due to being sick was analysed. However, absence from school was not associated with the daily amount of precipitation and wind speed.

This study has several limitations. First, the data of ambient air pollutants and meteorological factors in this study was obtained from one fixed monitoring station in Matsue City, which could not represent the total exposure to schoolchildren. Second, we were unable to calculate the personal exposure to ambient air pollutants and desert sand dust particles including ambient and indoor. Due to a lack of calculating the minute exposure to ambient air pollutants and desert sand dust particles in an individual, our study might underestimate or overestimate the effects of ambient air pollutants and desert sand dust particles on absence from school due to being sick. Third, being sick was based on a system of self-report and not confirmed by physicians and hence may include cases of feigned sickness. This study might overestimate the effects of ambient air pollutants and desert sand dust particles on absence from school due to being sick. Fourth, this study was lacking in data during long vacations such as summer and winter vacation. The concentration of O_x_ is highest in summer than in other seasons. Therefore, this study might underestimate the influence of O_x_ on absence from school due to being sick. Finally, this study was unable to assess the characteristics of the schoolchildren. A number of studies have found differences in the influence of ambient air pollutants on health between children with and without disease such as asthma. Prevalence rate various disease, especially respiratory disease, in schoolchildren influenced the outcome.

## 5. Conclusions

The present study evaluated the effects of PM_2.5_, NO_2_, and desert sand dust particles from East Asia on the overall absence of children from school due to sickness. Furthermore, this study observed positive associations of PM_2.5_, NO_2_, and desert sand dust particles with absence from school due to fever and cough. Healthy development during childhood is important for later life well-being, and sincere efforts for the reduction of air pollution are required to protect child health.

## Figures and Tables

**Figure 1 ijerph-18-06631-f001:**
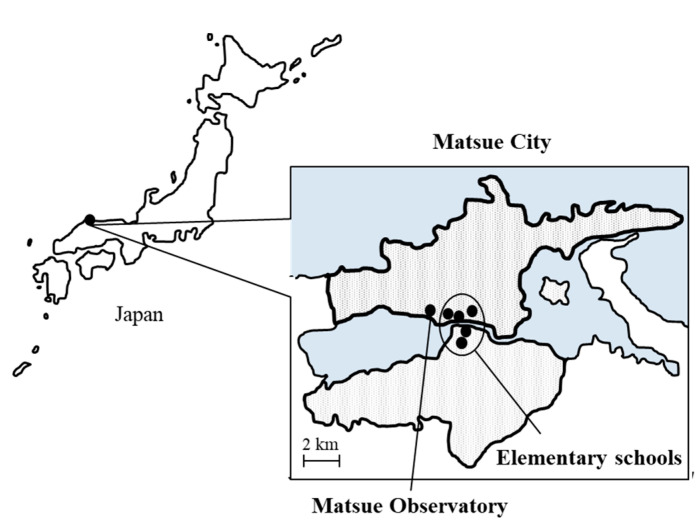
The spatial distributions of air monitoring sites and elementary schools in Matsue, Japan.

**Table 1 ijerph-18-06631-t001:** Descriptive statistics for meteorological factors, the concentration of air pollutants and desert sand dust particles by Light Detection and Ranging system from 1 November 2016 to 20 July 2018 in Matsue City, Japan.

Factors	Mean ± SD	Min	1st Quartile	Median	3rd Quartile	Max
Daily average temperature (°C)	14.2 ± 8.2	−2.6	7.1	13.8	21.2	32.3
Daily maximum temperature (°C)	18.6 ± 8.6	−0.3	11.2	18.7	26.1	37.7
Daily minimum temperature (°C)	10.3 ± 8.3	−7.0	3.0	9.0	16.9	28.3
Daily average relative humidity (%)	75.9 ± 10.4	43.0	69.0	76.0	83.0	98.0
Daily average atmospheric pressure (hPa)	1013.1 ± 6.7	992.9	1008.0	1013.4	1018.4	1027.9
Daily average PM_2.5_ (μg/m^3^)	12.4 ± 7.2	1.0	7.2	10.8	16.2	57.5
Daily maximum PM_2.5_ (μg/m^3^)	24.1 ± 15.8	2.0	13.0	21.0	30.0	134.0
Daily average SO_2_ (ppb)	1.0 ± 1.1	0	0.4	0.6	1.1	8.9
Daily maximum SO_2_ (ppb)	3.7 ± 5.7	0.1	0.7	1.2	4.0	47.6
Daily average NO_2_ (ppb)	2.3 ± 1.3	0.2	1.4	2.0	2.8	12.9
Daily maximum NO_2_ (ppb)	4.7 ± 3.0	0.6	2.8	3.9	5.8	32.1
Daily average O_x_ (ppb)	36.3 ± 13.0	0.8	28.8	36.5	44.1	90.6
Daily maximum O_x_ (ppb)	50.4 ± 16.7	0.17	41.7	48.6	58.7	120.6
Desert sand dust particles (km^−1^)	0.018 ± 0.022	0	0.008	0.012	0.019	0.233

SD, standard deviation; PM_2.5_, particulate matter less than 2.5 μm aerodynamic diameters; SO_2_, sulfur dioxide; NO_2_, nitrogen dioxide; and O_x_, photochemical oxidants.

**Table 2 ijerph-18-06631-t002:** The coefficient of spearman rank correlation between air pollutants and meteorological factors in Matsue City, Japan.

Factors	PM_2.5_	SO_2_	NO_2_	O_x_	DSDP	TEM	RHU	ATP
PM_2.5_	1							
SO_2_	0.26 *	1						
NO_2_	0.36 *	0.35 *	1					
O_x_	0.18 *	0.09 *	−0.02	1				
DSDP	0.49 *	0.23 *	0.23 *	0.48 *	1			
TEM	0.21 *	−0.26 *	−0.42 *	−0.09 *	−0.09 *	1		
RHU	−0.14 *	−0.15 *	0.17*	−0.44 *	−0.39 *	−0.01	1	
ATP	−0.04	0.16 *	0.38 *	−0.12 *	0.01	−0.66 *	−0.12 *	1

PM_2.5_, particulate matter less than 2.5 μm aerodynamic diameters; SO_2_, sulfur dioxide; NO_2_, nitrogen dioxide; and O_x_, photochemical oxidants; DSDP, desert sand dust particles; TEM, temperature; RHU, relative humidity; ATP, atmospheric pressure; * *p* < 0.05.

**Table 3 ijerph-18-06631-t003:** Descriptive statistics for daily absence cases from school in overall, fever and cough.

Factors	Total	Mean ± SD	Min	1st Quartile	Median	3rd Quartile	Max
Overall (*n*)	16,915	47.0 ± 37.3	4	26	33	51	258
Fever (*n*)	4,865	13.5 ± 7.2	0	9	12	16	52
Cough (*n*)	2,458	6.8 ± 4.5	0	4	6	9	35

SD, standard deviation.

**Table 4 ijerph-18-06631-t004:** Estimated relative risk and 95% confidence interval for overall absence from school due to sickness associated with a 10-µg/m^3^ increase in PM_2.5_, a 1-ppb in NO_2_, SO_2_ and O_x_ and a 0.1-km^−1^ in desert sand dust particles.

Lag days	PM_2.5_	SO_2_	NO_2_	O_x_	Desert Sand Dust Particles
Lag0	1.28 ^§^ (1.15, 1.42)	1.03 (0.99, 1.08)	1.03 (0.98, 1.08)	1.00 (0.99, 1.01)	2.15 * (1.04, 4.45)
Lag1	1.31 ^§^ (1.17, 1.46)	1.02 (0.97, 1.06)	1.02 (0.97, 1.07)	0.99 (0.98, 1.01)	2.46 ** (1.27, 4.79)
Lag2	1.32 ^§^ (1.18, 1.48)	0.95 (0.90, 1.01)	1.07 * (1.02, 1.13)	0.99 (0.98, 1.00)	1.29 (0.79, 2.10)
Lag3	1.23 ^§^ (1.11, 1.37)	0.96 (0.92, 1.01)	1.08 ^§^ (1.04, 1.13)	0.99 (0.98, 1.00)	1.71 * (1.03, 2.84)
Lag4	1.22 ^§^ (1.10, 1.36)	1.02 (0.97, 1.08)	1.04 * (1.00, 1.08)	1.00 (0.99, 1.01)	1.45 (0.84, 2.48)
Lag5	1.25 ^§^ (1.11, 1.40)	0.99 (0.95, 1.05)	1.03 (0.99, 1.07)	1.00 (0.99, 1.01)	1.57 (0.89, 2.76)

PM_2.5_, particulate matter less than 2.5 μm aerodynamic diameters; SO_2_, sulfur dioxide; NO_2_, nitrogen dioxide; and O_x_, photochemical oxidants; * *p* < 0.05; ** *p* < 0.01; ^§^
*p* < 0.001.

**Table 5 ijerph-18-06631-t005:** Estimated relative risk and 95% confidence interval for absence from school due to fever associated with a 10-µg/m^3^ increase in PM_2.5_, a 1-ppb in NO_2_, SO_2_ and O_x_ and a 0.1-km^−1^ in desert sand dust particles.

Lag Days	PM_2.5_	SO_2_	NO_2_	O_x_	Desert Sand Dust Particles
Lag0	1.20 ** (1.07, 1.36)	0.99 (0.94, 1.05)	1.04 (0.98, 1.10)	1.00 (0.99, 1.01)	2.92 ** (1.33, 6.41)
Lag1	1.22 ** (1.08, 1.38)	1.00 (0.95, 1.06)	1.03 (0.96, 1.09)	1.00 (0.98, 1.01)	3.32 ** (1.49, 7.38)
Lag2	1.25 ^§^ (1.11, 1.42)	0.95 (0.89, 1.02)	1.06 (0.99, 1.13)	0.99 (0.98, 1.01)	1.22 (0.71, 2.09)
Lag3	1.17 * (1.04, 1.33)	0.95 (0.89, 1.02)	1.09** (1.03, 1.15)	0.99 (0.98, 1.01)	1.31 (0.76, 2.29)
Lag4	1.18 * (1.04, 1.34)	1.04 (0.97, 1.11)	1.02 (0.96, 1.07)	1.01 (0.99, 1.02)	1.27 (0.66, 2.45)
Lag5	1.26 ** (1.09, 1.45)	0.96 (0.89, 1.03)	1.00 (0.94, 1.06)	1.01 (0.99, 1.02)	1.41 (0.70, 2.85)

PM_2.5_, particulate matter less than 2.5 μm aerodynamic diameters; SO_2_, sulfur dioxide; NO_2_, nitrogen dioxide; and O_x_, photochemical oxidants; * *p* < 0.05; ** *p* < 0.01; ^§^
*p* < 0.001.

**Table 6 ijerph-18-06631-t006:** Estimated relative risk and 95% confidence interval for absence from school due to cough associated with a 10-µg/m^3^ increase in PM_2.5_, a 1-ppb in NO_2_, SO_2_ and O_x_ and a 0.1-km^−1^ in desert sand dust particles.

Lag Days	PM_2.5_	SO_2_	NO_2_	O_x_	Desert Sand Dust Particles
Lag0	1.18 * (1.01, 1.37)	1.06 * (1.01, 1.13)	1.06 (0.99, 1.13)	1.01 (0.99, 1.02)	2.70 * (1.05, 6.94)
Lag1	1.18 * (1.01, 1.38)	1.03 (0.97, 1.10)	1.02 (0.95, 1.10)	1.01 (0.99, 1.02)	2.07 (0.77, 5.54)
Lag2	1.08 (0.92, 1.25)	1.03 (0.96, 1.11)	1.03 (0.96, 1.11)	1.01 (0.99, 1.02)	0.97 (0.45, 2.12)
Lag3	1.02 (0.88, 1.18)	0.98 (0.91, 1.06)	1.09 * (1.02, 1.16)	1.00 (0.99, 1.02)	1.34 (0.65, 2.75)
Lag4	1.17 (1.00, 1.37)	0.97 (0.88, 1.06)	1.12 ** (1.04, 1.20)	1.01 (0.99, 1.02)	0.87 (0.41, 1.85)
Lag5	1.10 (0.93, 1.30)	0.99 (0.90, 1.08)	1.13 ^§^ (1.05, 1.22)	1.01 (0.99, 1.02)	0.82 (0.34, 2.00)

PM_2.5_, particulate matter less than 2.5 μm aerodynamic diameters; SO_2_, sulfur dioxide; NO_2_, nitrogen dioxide; and O_x_, photochemical oxidants; * *p* < 0.05; ** *p* < 0.01; ^§^
*p* < 0.001.

**Table 7 ijerph-18-06631-t007:** Estimated relative risk and 95% confidence interval for absence associated with a 10-µg/m^3^ increase in PM_2.5_, a 1-ppb in NO_2_, SO_2_ and O_x_ and a 0.1-km^−1^ in desert sand dust particles at lag1 in two-pollutant models.

Variable	PM_2.5_	O_x_	NO_2_	SO_2_	Desert Sand Dust Particles
Over all absence					
Adjusted for PM _2.5_	−	1.00 (0.99, 1.00)	0.98 (0.94, 1.04)	1.01 (0.97, 1.05)	1.84 (0.97, 3.47)
Adjusted for O_x_	1.30 (1.17, 1.45)	−	1.00 (0.94, 1.07)	1.03 (0.98, 1.08)	2.66 (1.37, 5.17)
Adjusted for NO_2_	1.32 (1.17, 1.50)	0.99 (0.98, 1.01)	−	1.02 (0.97, 1.07)	2.30 (1.17, 4.52)
Adjusted for SO_2_	1.30 (1.16, 1.45)	0.99 (0.98, 1.00)	1.02 (0.97, 1.08)	−	2.39 (1.25, 4.57)
Adjusted for desert sand dust particles	1.30 (1.12, 1.51)	0.99 (0.98, 1.00)	1.10 (1.03, 1.19)	1.04 (0.97, 1.11)	−
Absence due to fever					
Adjusted for PM _2.5_	−	1.00 (0.99, 1.01)	1.00 (0.94, 1.07)	1.00 (0.94, 1.05)	2.74 (1.21, 6.18)
Adjusted for O_x_	1.22 (1.08, 1.38)	−	1.02 (0.95, 1.09)	1.01 (0.96, 1.07)	3.67 (1.62, 8.32)
Adjusted for NO_2_	1.19 (1.03, 1.37)	1.00 (0.98, 1.01)	−	1.01 (0.95, 1.07)	2.59 (1.12, 5.99)
Adjusted for SO_2_	1.22 (1.08, 1.39)	1.00 (0.99, 1.01)	1.03 (0.96, 1.10)	−	3.43 (1.49, 7.90)
Adjusted for desert sand dust particles	1.15 (0.98, 1.36)	0.99 (0.98, 1.00)	1.09 (1.01, 1.19)	0.99 (0.92, 1.07)	−
Absence due to cough					
Adjusted for PM _2.5_	−	1.01 (0.94, 1.02)	1.01 (0.93, 1.09)	1.03 (0.97, 1.09)	1.63 (0.63, 4.27)
Adjusted for O_x_	1.19 (1.01, 1.39)	−	1.06 (0.97, 1.16)	1.03 (0.96, 1.09)	2.07 (0.76, 5.64)
Adjusted for NO_2_	1.16 (0.97, 1.39)	1.02 (1.00, 1.03)	−	1.03 (0.97, 1.10)	2.28 (0.79, 6.60)
Adjusted for SO_2_	1.16 (0.99, 1.36)	1.00 (0.99, 1.02)	1.03 (0.96, 1.11)	−	1.80 (0.67, 4.85)
Adjusted for desert sand dust particles	1.31 (1.05, 1.65)	1.000 (0.98, 1.02)	1.03 (0.92, 1.15)	1.05 (0.96, 1.15)	−

PM_2.5_, particulate matter less than 2.5 μm aerodynamic diameters; SO_2_, sulfur dioxide; NO_2_, nitrogen dioxide; and O_x_, photochemical oxidants.

**Table 8 ijerph-18-06631-t008:** Estimated relative risk and 95% confidence interval for overall absence from school due to sickness associated with a 10 °C increase in daily average of temperature, a 10 % in relative humidity, and a 10 hPa in atmospheric pressure.

Lag Days	Temperature (°C)	Relative Humidity (%)	Atmospheric Pressure (hPa)
Lag0	0.86 (0.67, 1.10)	1.03 (0.97, 1.10)	0.96 (0.86, 1.08)
Lag1	0.89 (0.69, 1.13)	1.00 (0.95, 1.06)	0.92 (0.81, 1.04)
Lag2	0.89 (0.70, 1.13)	0.97 (0.92, 1.02)	0.98 (0.86, 1.12)
Lag3	0.88 (0.69, 1.13)	0.95 (0.90, 1.01)	1.21 * (1.05, 1.38)
Lag4	0.72 ** (0.56, 0.92)	1.01 (0.96, 1.07)	1.22 * (1.08, 1.38)
Lag5	0.71 ** (0.56, 0.91)	1.08 * (1.01, 1.15)	1.08 (0.97, 1.21)

* *p* < 0.05; ** *p* < 0.01.

## Data Availability

The datasets used and/or analyzed during the current study are available from the corresponding author on reasonable request.

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
