# Peer review of "Association with Ambient Air Pollutants and School Absence Due to Sickness in Schoolchildren: A Case-Crossover Study in a Provincial Town of Japan"

_ijerph, 2021, doi:10.3390/ijerph18126631_

Round 1
Reviewer 1 Report
This study investigated the influence of ambient air pollutants and desert sand dust from East Asia on absence from school due to sickness by conducting a case-control study. This paper shows an interesting topic. Some minor revisions are needed.
1. The last sentence of the abstract says, “increased the risk of absence from due to sickness in schoolchildren”. What does it mean by “from due to”? “ambient air pollutant” should be written by “ambient air pollutants”. “a case control study” should be written by “a case-control study”. Please check the grammar carefully in the whole paper.
2. At the end of the Introduction, the contribution and innovation of this research should be pointed out directly.
3. There is no need to keep China, North Korea, and South Korea in Figure 1. Make sure to keep Japan in Figure 1 only.
4. In the Introduction, some more papers need to be added to emphasize the importance of this research. In the discussion part, please compare this study with previous studies such as:
Environmental regulations on air pollution in China and their impact on infant mortality. https://doi.org/10.1016/j.jhealeco.2015.02.004
Spatial Heterogeneity Influences of Environmental Control and Informal Regulation on Air Pollutant Emissions in China. https://doi.org/10.3390/ijerph17134857
5. In the Conclusion part, it would be interesting to focus on making a critical assessment around the generalization of the results.
Author Response
Manuscript Number: IJERPH-1249430
Point-by-point responses to the reviewer’s comments
Reviewer 1
Comment 1: The last sentence of the abstract says, “increased the risk of absence from due to sickness in schoolchildren”. What does it mean by “from due to”? “ambient air pollutant” should be written by “ambient air pollutants”. “a case control study” should be written by “a case-control study”. Please check the grammar carefully in the whole paper.
Response: We have checked and corrected grammar in our manuscript. It was edited again by a native at Editage (www.editage.jp).
Comment 2: At the end of the Introduction, the contribution and innovation of this research should be pointed out directly.
Response: Thank you for your suggestion. We have clarified the contribution and innovation of this research in lines 68–75.
Comment 3: There is no need to keep China, North Korea, and South Korea in Figure 1. Make sure to keep Japan in Figure 1 only.
Response: We have corrected Figure 1.
Comment 4: In the Introduction, some more papers need to be added to emphasize the importance of this research. In the discussion part, please compare this study with previous studies such as:
Environmental regulations on air pollution in China and their impact on infant mortality. https://doi.org/10.1016/j.jhealeco.2015.02.004
Spatial Heterogeneity Influences of Environmental Control and Informal Regulation on Air Pollutant Emissions in China. https://doi.org/10.3390/ijerph17134857
Response: Thank you for your suggestion. We may be unable to fully understand your intention. To our best effort, we have added a discussion comparing previous studies, including your recommendations, in lines 284–300. We also added sentences and references in the Introduction section.
Comment 5: In the Conclusion part, it would be interesting to focus on making a critical assessment around the generalization of the results.
Response: We have corrected the Conclusion section as follows: The present study evaluated the effects of PM2.5, NO2, and desert sand dust particles on the overall absence of children from school due to being sick. Furthermore, this study observed positive associations between PM2.5, NO2, and desert sand dust particles and absence from school due to fever and cough. Healthy development during childhood is important for later life well-being, and we may have to take serious measures for the reduction of air pollution to protect child health.

Reviewer 2 Report
This is a straightforward description of an ecological study of school absence in relation to air pollutants. The authors are careful not to overstate the implications in terms of causation, which is entirely appropriate. However, I feel that some further discussion of the problems of inferring causality in this type of study is warranted and the conclusions should state that the study cannot confirm that PM2.5, NO2, or desert sand dust particles caused absence from school due to fever or cough.
The text could be improved with minor editing to improve the English language.
Minor comments
Line 43: “NO2 and SO2 should have the numbers subscripted (also at other points in the text super / subscription must be corrected, e.g. line 112, 113, 114…)
Line 43: “or a numbers of health disorder”. English and please explain and provide a citation to substantiate.
Line 49: “…and can attribute deaths” English
Line 84: Appears to be an incomplete sentence.
Table 2: What is DDA?
Table 3: Define Mean (as “daily mean absence cases”).
Table 4: Be consistent in format of units, e.g. μg/m3 but elsewhere have used km-1
Table 7: two decimal places to be consistent with earlier tables (3DP not justified)
Line 229: “was the largest risk factor in absence from school due to sickness” – However, these are measured in different units and so it is difficult to justify AD as the largest risk.
Line 243: “The present study also surveyed the number of cases of absence due to abdominal 243 symptoms, vomiting, rash, influenza infection, influenza like symptom.” Why are these data excluded? Why refer to the other causes of absence if they are not analysed.
Line 266: “On the contently, there were significant…” English
Author Response
Manuscript ID: IJERPH-1249430
Point-by-point response to the reviewer’s comments
Reviewer 2
Comment: Line 43: “NO2 and SO2 should have the numbers subscripted (also at other points in the text super / subscription must be corrected, e.g., line 112, 113, 114…)
Response: We carefully checked the super / subscription in the manuscript and used subscripted numbers.
Comment: Line 43: “or a numbers of health disorder”. English and please explain and provide a citation to substantiate.
Response: We have corrected the sentence and added references.
Comment: Line 49: “…and can attribute deaths” English
Response: We have corrected the sentence.
Comment: Line 84: Appears to be an incomplete sentence.
Response: We have corrected the sentences as follows: The hourly concentrations of PM2.5, SO2, NO2, and photochemical oxidants (Ox) are monitored by the Japanese Ministry of the Environment in Matsue City, and these data were obtained from their database. The data of meteorological variables, including daily temperature, humidity, and atmospheric pressure, were obtained from the Japan Meteorological Agency. Line 94.
Comment: Table 2: What is DDA?
Response: We have revised DDA to DSDP.
Comment: Table 3: Define Mean (as “daily mean absence cases”).
Response: We have corrected the title in Table 3.
Comment: Table 4: Be consistent in format of units, e.g. μg/m3 but elsewhere have used km-1
Response: The degree of each parameter, such as PM2.5, SO2, NO2, Ox, and desert sand dust particles, is calculated by a different unit. Therefore, we apologise the inconsistency in the format of units. We have already provided an explanation about a unit of desert sand dust particle levels in the Materials and methods section, 118–123.
Comment: Table 7: two decimal places to be consistent with earlier tables (3DP not justified)
Response: We have corrected from three decimal places to 2DP.
Comment: Line 229: “was the largest risk factor in absence from school due to sickness” – However, these are measured in different units and so it is difficult to justify AD as the largest risk.
Response: We have deleted the following sentence: Additionally, desert sand dust particles, among all the air pollutions, was the largest risk factor in absence from school due to sickness.
Comment: Line 243: “The present study also surveyed the number of cases of absence due to abdominal symptoms, vomiting, rash, influenza infection, influenza like symptom.” Why are these data excluded? Why refer to the other causes of absence if they are not analysed.
Response: Thank you for your suggestion. The classifications of abdominal symptoms, vomiting, rash, influenza infection, and others were included to decrease the efforts required by the schoolteachers based on their routine survey. We did not want these data because this study focused on the influence of infection and respiratory disease exposure to ambient air pollution and desert sand dust on absence from school. We have added an explanation in study design, Material and Methods section, lines 78–88, and corrected the sentence in Discussion, line 302–308.
Comment: Line 266: “On the contently, there were significant…” English
Response: We have corrected as follows from lines 322–330: Therefore, average temperature, relative humidity, and atmospheric pressure were included in the model to control the confounding effects of meteorological factors on the association between air pollution, which included desert sand dust particles and absence from school. Moreover, the present study also analysed the effects of the meteorological factors on absence from school and found statistical significances. PM2.5 and sand dust particles were significantly associated with absence from school in lag0, lag 1, and lag2 but the significant relative risk for meteorological factors on absence from school lacked in lag0, lag 1, and lag2. Meteorological factors did not affect interactions between ambient air pollutants, desert sand dust particles, and absence from school due to sickness.